# Short-Interval, Low-Dose Peptide Receptor Radionuclide Therapy in Combination with PD-1 Checkpoint Immunotherapy Induces Remission in Immunocompromised Patients with Metastatic Merkel Cell Carcinoma

**DOI:** 10.3390/pharmaceutics14071466

**Published:** 2022-07-14

**Authors:** Alexandra Aicher, Anca Sindrilaru, Diana Crisan, Wolfgang Thaiss, Jochen Steinacker, Meinrad Beer, Thomas Wiegel, Karin Scharffetter-Kochanek, Ambros J. Beer, Vikas Prasad

**Affiliations:** 1Department of Dermatology and Allergology, University of Ulm, 89081 Ulm, Germany; aicher_a@yahoo.com (A.A.); mihaela-anca.sindrilaru@uniklinik-ulm.de (A.S.); diana.crisan@uniklinik-ulm.de (D.C.); karin.scharffetter-kochanek@uniklinik-ulm.de (K.S.-K.); 2Precision Immunotherapy Group, Graduate Institute of Biomedical Sciences, China Medical University, Taichung 404333, Taiwan; 3Department of Nuclear Medicine, University of Ulm, 89081 Ulm, Germany; wolfgang.thaiss@uniklinik-ulm.de (W.T.); jochen.steinacker@uniklinik-ulm.de (J.S.); ambros.beer@uniklinik-ulm.de (A.J.B.); 4Department of Diagnostic and Interventional Radiology, University of Ulm, 89081 Ulm, Germany; meinrad.beer@uniklinik-ulm.de; 5Department of Radiotherapy and Radio-Oncology, University of Ulm, 89081 Ulm, Germany; thomas.wiegel@uniklinik-ulm.de

**Keywords:** radiotherapy, immunotherapy, checkpoint inhibitors, anti-PD-1, neuroendocrine tumors

## Abstract

Merkel cell carcinoma (MCC) is a neuroendocrine skin cancer of the elderly, with high metastatic potential and poor prognosis. In particular, the primary resistance to immune checkpoint inhibitors (ICI) in metastatic (m)MCC patients represents a challenge not yet met by any efficient treatment modality. Herein, we describe a novel therapeutic concept with short-interval, low-dose ^177^Lutetium (Lu)-high affinity (HA)-DOTATATE [^177^Lu]Lu-HA-DOTATATE peptide receptor radionuclide therapy (SILD-PRRT) in combination with PD-1 ICI to induce remission in patients with ICI-resistant mMCC. We report on the initial refractory response of two immunocompromised mMCC patients to the PD-L1 inhibitor avelumab. After confirming the expression of somatostatin receptors (SSTR) on tumor cells by [^68^Ga]Ga-HA-DOTATATE-PET/CT (PET/CT), we employed low-dose PRRT (up to six treatments, mean activity 3.5 GBq per cycle) at 3–6 weeks intervals in combination with the PD-1 inhibitor pembrolizumab to restore responsiveness to ICI. This combination enabled the synergistic application of PD-1 checkpoint immunotherapy with low-dose PRRT at more frequent intervals, and was very well tolerated by both patients. PET/CTs demonstrated remarkable responses at all metastatic sites (lymph nodes, distant skin, and bones), which were maintained for 3.6 and 4.8 months, respectively. Both patients eventually succumbed with progressive disease after 7.7 and 8 months, respectively, from the start of treatment with SILD-PRRT and pembrolizumab. We demonstrate that SILD-PRRT in combination with pembrolizumab is safe and well-tolerated, even in elderly, immunocompromised mMCC patients. The restoration of clinical responses in ICI-refractory patients as proposed here could potentially be used not only for patients with mMCC, but many other cancer types currently treated with PD-1/PD-L1 inhibitors.

## 1. Introduction

Metastatic Merkel cell carcinoma (mMCC) is an aggressive skin cancer of neuroendocrine origin, with a poor 5-year overall survival (OS) rate of <20% [1]. Typically, MCC occurs in UV-damaged skin in elderly, immunosuppressed patients [2]. The incidence of MCC is increasing worldwide, most likely driven by a gradual decline in the immune defense of an ageing population [3]. MCC is highly immunogenic, due to the Merkel cell polyoma virus (MCPyV) clonally integrated into the host genome in about 80% of cases, and the heavy UV exposure resulting in a distinct UV-damage signature with high mutational burden [4,5]. Therefore, MCC is a promising candidate for immunotherapy using PD-L1 or PD-1 immune checkpoint inhibition (ICI). Consequently, the PD-L1 inhibitor avelumab received regulatory approval as first-line therapy in mMCC in both the United States and Europe in 2017 [6], and the PD-1 inhibitor pembrolizumab is currently also approved for mMCC therapy in the United States. However, primary and secondary resistance to immunotherapy occurs in about 50% of mMCC patients treated with ICI, and underscores the unmet need for novel therapeutic strategies in these patients [7].

Conventional radiotherapy significantly improves relapse-free survival in adjuvant settings [8]. However, palliative radiation of MCC metastases is associated with poor distant metastasis-free survival [9]. A growing body of evidence suggests that external beam radiotherapy (EBRT), particularly when applied according to hypofractionated protocols as three single individual fractions of 5 Gy to 8 Gy, may sensitize the tumor in refractory patients to become responsive to re-challenging with ICI [10,11]. Thus, different total therapeutic doses and dose rates result in significantly different gene expression patterns and, therefore, may decisively modulate, and even restore, responsiveness to ICI [12]. Irrespective of the radiation protocol, palliative EBRT can only be used for single or very few metastases. Here, we employed internal radiation therapy to simultaneously target multiple metastatic lesions. To overcome this limitation, we intravenously administered [^177^Lu]Lu-HA-DOTATATE, a radioconjugate of the peptide-based somatostatin analogue 3-iodo-Tyr(3),Thr(8)octreotide labeled with Lutetium 177 (^177^Lu) chelated with DOTA, for peptide receptor radionuclide therapy (PRRT) [13]. [^177^Lu]Lu-DOTATATE delivers ionizing radiation mainly as beta particle emission to cells expressing somatostatin receptors (SSTR), which are overexpressed in neuroendocrine tumors such as MCC [14]. One single dose of 7.4 GBq [^177^Lu]Lu-DOTATATE was shown to re-sensitize cells to anti-PD-L1 immunotherapy, however, long-term follow-up data are missing [15]. Currently, the recommended PRRT schedule is 7.4 GBq provided in four cycles every 8–10 weeks. Due to the aggressive behavior of mMCC and the high relapse rate to ICI, we propose a novel concept with low-dose PRRT (3.5 GBq) administered every 3–6 weeks concomitantly with anti-PD-1 treatment (Figure 1).

## 2. Material and Methods

### 2.1. Radiopharmaceutical Preparation, PRRT and Imaging

The ^177^Lu was acquired from ITM (Garching, Germany). HA-DOTATATE was purchased from Scintomics (Munich, Germany). Ga-68 was eluted from an iThemba Ge-68/Ga-68 Generator. Radiolabelling with ^68^Ga and ^177^Lu was performed in accordance with procedures in GMP-certified radiopharmacy laboratories, as previously described [16]. [^177^Lu]Lu-HA-DOTATATE and amino acids were employed according to the ENETS consensus guidelines, including the joint IAEA, EANM, and SNMMI guidelines [16,17,18]. ^177^Lu]Lu-HA-DOTATATE and [^68^Ga]Ga-HA-DOTATATE were prepared in compliance with the German Medicinal Product Act, §13.2b. Both radiopharmaceuticals have been regularly used in clinical practice in Germany for more than 10 years [19]. [^68^Ga]Ga-HA-DOTATATE-PET/CT (PET/CT) was performed pre- and post-PRRT. Pembrolizumab (fixed dose of 200 mg) was regularly given every three weeks. PRRT was administered every 3 to 6 weeks concomitantly. PRRT and pembrolizumab were never given within the same week, and there was a gap of at least 7 days between the two therapies. Schematics were created with BioRender.com accessed on 13 May 2022.

### 2.2. Ethical Approval

Therapeutic decisions were made by consensus at regular multidisciplinary meetings. The tumor board of the dermatological oncology department in the University of Ulm consists of dermatologists including dermatologists from other independent hospitals, oncologists, radiation oncologists, radiologists, and nuclear medicine physicians. We obtained written informed consent from the patients for all performed procedures. The radiopharmaceuticals were produced in accordance with the German Pharmaceuticals Act (§13.2b) and the responsible regulatory authorities. The retrospective evaluation of data was approved by the ethics committee of the University of Ulm (181/21) on 2 June 2021.

## 3. Case Description

*Patient A*. An 82-year-old male with a history of B-cell chronic lymphatic leukemia (B-CLL) since 2012, achieving stable partial remission after four cycles of obinutuzumab/chlorambucil in 2015. In December 2019, he was diagnosed with MCC in his right elbow, with multiple satellite lesions and inoperable regional axillary lymph node macrometastases (pT3pN1cM0). In January 2020, the patient started a first-line immunotherapy with avelumab 800 mg q2w in combination with local EBRT of the primary tumor region and the axillary lymph nodes (45 Gy; 3 Gy/fraction), with an initial partial response in the irradiated regions but not the distant metastases. He relapsed early after radiotherapy with new skin metastases of the right arm. Despite changing to pembrolizumab 200 mg q3w in parallel to EBRT of the forearm (45 Gy; 3 Gy/fraction), and a single EBRT (8 Gy) of further in-transit skin metastases on the lower arm (April 2020– June 2020), PET/CT images from July 2020 reveal disease progression with disseminated bone metastases of the cervical and thoracic spine, ischial tuberosities, and the proximal left femur, strongly suggesting resistance to a PD-1/PD-L1 pathway blockade. Based on moderate to intensive SSTR expression of metastases in the PET/CT-scan, we initiated SILD-PRRT in combination with pembrolizumab immunotherapy. After two cycles of intravenously administered [^177^Lu]Lu-HA-DOTATATE, with a mean activity of 3.5 GBq (range 3–4.2 GBq) and concomitant pembrolizumab, the fused axial PET/CT scans of 09/2020 show an almost complete response of all osseous metastases that lasted until November 2020 (Figure 2 and Figure 3). However, in December 2020, after four cycles of SILD-PRRT with pembrolizumab and 4.8 months after the first PRRT, he relapsed with the progression of skin and lymph node metastases not responding any further to combined SILD-PRRT–pembrolizumab therapy. The scan of February 2021 shows multifocal disease progression with soft tissue, skin, and lymph node metastases of the right limb and axilla, and new diffused bone involvement of the thoracic skeletal system. Patient A succumbed to disease 8 months after initiating PRRT, after receiving and tolerating very well six SILD-PRRT treatments with ongoing ICI.

*Patient B*. An 85-year-old female with a history of B-CLL since 2009. The patient was diagnosed with MCC of her left nostril (pT1 cN0 cM0) in 2014. In 01/2020, she progressed with rapidly growing distant skin/soft-tissue metastases of the right lower limb and right inguinal lymph node metastases. In January 2020, we initiated systemic immunotherapy with avelumab, combined with palliative EBRT of the skin lesions (45 Gy; 2.5 Gy/fraction). In June 2020, Patient B relapsed with new skin metastases on her right lower limb, with high SSTR expression in the PET/CT-scan. In June 2020, the patient received pembrolizumab in combination with SILD-PRRT. After three cycles of SILD-PRRT q4w (Figure 4 and Figure 5), we found a substantially lower uptake of ^177^Lu compared with the initial PET/CT scans of the right leg, and microscopic examination of a skin metastasis revealed complete necrotic tumor tissue. However, the patient progressed with skin and lymph node metastases involving inguinal and intraabdominal lymph nodes 3.6 months after the first PRRT (October 2020). She died from renal failure 7.7 months following PRRT–pembrolizumab initiation. Overall, she received three SILD-PRRT treatments with concomitant pembrolizumab, with a mean activity of 3.5 GBq/cycle at 4-week intervals.

### 3.1. Integral Dose Assessment

We determined time–activity curves in different regions of interest (ROI) images in patient A. Using physiologically based pharmacokinetic modeling for dosimetry, we estimated the integral Gy per MBq delivered to the tumor for each PRRT (Figure 6). ROI images laid over the bone metastasis in the left femur and cervical vertebrae indicate that only 4.7 Gy and 5.4 Gy, respectively, are delivered to the bone metastases, while ROI images of the right elbow demonstrate the delivery of 25 Gy to the skin tumor. Intriguingly, the patient’s skeletal metastases responded very well to the treatment and remained tumor-free, while we could not completely eradicate the skin tumor in the elbow region, most probably due to previous EBRT with 45 Gy. In fact, disease relapsed at this site later in November 2020.

### 3.2. Toxicity Assessment

As expected, lymphocyte counts decreased for both patients after the induction of PRRT, whereas neutrophil counts remained within the normal range (Figure 7). Eosinophils were counted to assess the response to treatment, as eosinophilia is suggested as a prognostic marker for the response in patients with metastatic melanoma following ICI treatment [20]. Interestingly, eosinophilia in patient A consistently correlated with the response to pembrolizumab and PRRT therapy, whereas eosinophil counts fell with tumor relapse. Leukocyte, erythrocyte, and thrombocyte counts were not affected after the initial PRRT. Hemoglobin and creatinine concentrations did not markedly change with respect to baseline levels in June 2020 prior to the initial PRRT (data not shown).

Furthermore, in both patients, we detected Merkel cell polyomavirus by polymerase chain reaction (PCR).

## 4. Discussion

In these two single cases, we evaluated the feasibility of combining SILD-PRRT with the PD-1 inhibitor pembrolizumab. The results suggest that this combination has the potential to restore responsiveness in patients with mMCC that are refractory to first-line PD-L1 ICI monotherapy. Although both patients were immunosuppressed due to CLL, and primarily resistant to avelumab, we observed a partial remission of bone metastases in patient A, and a partial remission of extensive distant skin metastases in patient B using PET/CT. Both patients tolerated the combination therapy very well. To our knowledge, this is the first report employing a modified PRRT regimen concomitantly with a PD-1 inhibitor in immunocompromised mMCC patients.

The rationale to employ PRRT in combination with PD-1 inhibition was to overcome resistance to PD-1/PD-L1 ICI [21]. New ICI therapies provide impressive results for many cancers including high objective response (OR) rates for mMCC. However, recent ‘real-life’ studies on patients who did not respond to immunotherapy show a dramatically poor prognosis, with a median PFS of 1.4 months and median OS of 3.9 months [22]. Notably, it is identified that immunosuppression strongly correlates with primary immune resistance. To date, such patients who previously received immunotherapy, or had immunosuppression, were excluded in clinical trials. Data available from small case series show virtually no response to second-line (radio)chemotherapy, or PD-1 ICI monotherapy, for patients having progressed with avelumab [23]. However, the immunocombination of ipilimumab and nivolumab shows some activity, but only in immunocompetent patients [23,24]. Furthermore, additional administration of conventional PRRT along with ipilimumab and nivolumab induced remission in an immunocompetent patient previously resistant to immunotherapy, albeit with severe, ipilimumab-specific toxicity [25]. As immune checkpoint inhibitors are currently tested for an overwhelming number of oncologic indications, SILD-PRRT could probably be combined with every other ICI in various types of cancer.

The mechanisms of primary and acquired immune resistance are complex and still need to be elucidated, but immunosuppression seems to be critically involved [22]. Surprisingly, tumor mutational burden, the presence of MCPyV, and PD-1/PD-L1 expression do not correlate with therapy response [22]. Here, we aimed at restoring and maintaining responsiveness to ICI by delivering low-dose radiation using PRRT at short intervals at metastatic sites. Notably, low-dose radiation results in reduced efficacy when administered as monotherapy in other neuroendocrine tumors [26]. The rationale for administering lower radioactivity (approximately 50% of standard dose) twice as frequently as compared to conventional regimens was to achieve radiation doses of 2–10 Gy in the tumor. This dose range was suggested to increase immunogenic, radiation-induced, double-stranded DNA, to upregulate HLA-I molecules and tumor antigen presentation, and favor M1 macrophage polarization and NK cell infiltration, thus, restoring responses to immunotherapy [27,28]. Indeed, we delivered an estimated dose of 4.7 Gy to areas of bone metastasis that responded very well to the combined treatment.

The systemic delivery to potentially all of the metastatic sites constitutes another substantial advantage of SILD-PRRT over EBRT. Thus, all SSTR positive lesions will convert into immunogenic or ‘hot’ tumors, with enhanced infiltration of cytotoxic CD8^+^ T cells and better response rates to ICI [29,30,31]. ‘Hot’ tumors are characterized by a type I interferon (IFN) signature. Type I IFN is the downstream product of a cascade of events sensing cytoplasmic double-stranded DNA (dsDNA), resulting from radiation-induced DNA damage. Eventually, type I IFN signaling can restore class I HLA expression that is essential to anti-cancer CD8^+^ T-cell responses.

Of note, the strategy outlined here for the treatment of patients suffering from mMCC was successfully applied, combining targeted radionuclide therapy using Y-90 alkylphosphocholine and ICI in a murine melanoma model [32]. This study clearly provides evidence for the superiority of the combinational treatment of targeted radionuclide therapy and ICI over each single approach. Notably, the somatostatin analogue octreotide utilized in our PRRT approach may also add to the favorable response, as octreotide administered via intramuscular injection every 28 days in combination with avelumab demonstrates durable disease control in a patient with mMCC [33]. SSTR2 are not only expressed by neuroendocrine neoplastic cells, but also by some tumor-infiltrating B-cells. Hence, direct anti-proliferative effects of octreotide on both neoplastic and immune cells of the tumor microenvironment are possible [34]. On the other hand, it is tempting to speculate that, by targeting neoplastic lymphocytes, PRRT may have coincidently improved the immunocompetence status of our CLL patients, as the peripheral lymphocyte counts of both patients substantially decreased during treatment, and neither patient showed CLL exacerbation. However, we did not perform further hematologic immunophenotyping on lymphocytes for our patients. Accordingly, ^177^Lu-DOTATATE was recently suggested as a cytoreductive therapy in chronic B-CLL [35].

Both patients tolerated the combination therapy very well. The concept of using SILD allowed us to better synchronize PRRT and ICI treatment to achieve synergistic responses. In addition, we rapidly obtained objective responses with more frequently administered low-dose PRRT and with PD-1 ICI monotherapy, thus, avoiding the severe toxicity of other potential immunotherapy combinations. Importantly, the protocols of both patients are largely consistent regarding pembrolizumab doses and schedule (200 mg every 3 weeks), including the mean activity of 3.5 GBq for each cycle of PRRT. The intervals for PRRT could not be strictly kept at a fixed interval, due to patient-related issues. However, we maintained the application of PRRT within 3–6 weeks intervals, in line with our SILD-PRRT concept.

Apart from the scientific evidence, this dose adaptation was necessary, as there are no data on the safety or tolerability of PRRT in combination with pembrolizumab in elderly (>80 years) patients. However, nephrotoxicity may occur after PRRT monotherapy. As ICI may also result in renal injury as an underestimated severe autoimmune-mediated adverse reaction, our regimen using lower doses for PRRT could, at least in our patients, avoid synergistic renal toxicity [36]. Thus, PRRT combined with ICI was tolerated well, with no signs of renal toxicity or generalized hematotoxicity. However, we cannot completely rule out whether the reported increase in efficacy observed following SILD-PRRT in combination with pembrolizumab might have been caused by PRRT monotherapy [37,38].

SILD-PRRT along with pembrolizumab as second-line therapy in two immunosuppressed elderly patients with primary immunoresistant mMCC was safe, well-tolerated, and resulted in a promising PFS of 4.8 and 3.6 months and OS of 8 and 7.7 months, respectively, representing a substantial improvement on currently available data [22,23]. The major limitations of this report were the small number of treated patients, and the lack of tissue material prior and after SILD-PRRT to prove that beta particle radiation restored the response to treatment by inducing favorable changes in the tumor microenvironment. An ongoing clinical trial (NCT04261855) may provide mechanistic insights, and also further explore the efficacy and safety of avelumab in combination with conventional PRRT and EBRT. Our data will initiate further investigations addressed at defining the potential of SILD-PRRT in other SSTR-expressing cancers. Following these two single patients with mMCC, we aim at exploring SILD-PRRT in combination with pembrolizumab by initiating a phase I clinical trial to assess the responsiveness to pembrolizumab in a larger number of mMCC patients. In future studies, the clinical and molecular responses of the tumor microenvironment to graded low-dose SILD-PRRT treatment could be prospectively monitored. Pre- and post-SILD-PRRT biopsies will provide insights into the molecular mechanisms to explain differences in the response of osseous versus skin and lymph node metastases to SILD-PRRT. Single-cell multiplex imaging of immunosuppressive immune cells neighboring SSTR2^+^ tumor cells will facilitate advances in understanding the complex radioligand–tumor cell interactions within the tumor microenvironment. Eventually, both patients became refractory to SILD-PRRT. To overcome resistance, more potent alpha-emitters may be superior for PRRT. SILD-PRRT is attractive to remodel the TME for future SSTR2-targeting therapies, such as bispecific T-cell engagers (for instance BiTEs for SSTR2 and CD3; NCT03411915), or SSTR2-targeting CAR-T cells.

## 5. Conclusions

We demonstrate that SILD-PRRT in combination with pembrolizumab is safe, and well-tolerated, even in elderly, immunocompromised mMCC patients, and has the potential to induce objective clinical responses. Both patients tolerated the therapy very well, and achieved overall remissions with considerably improved survival rates as compared to currently available data. The mechanisms underlying this outcome are still to be elucidated, but collectively, the choice of treatments, dosing, and regimen seem to contribute to restoring immune responsiveness in this setting. Future clinical trials are needed to validate this therapeutic concept for maximizing the synergistic role of PRRT and ICI in patients with mMCC and, in addition, many other ICI-refractory cancer types.

## Figures and Tables

**Figure 1 pharmaceutics-14-01466-f001:**
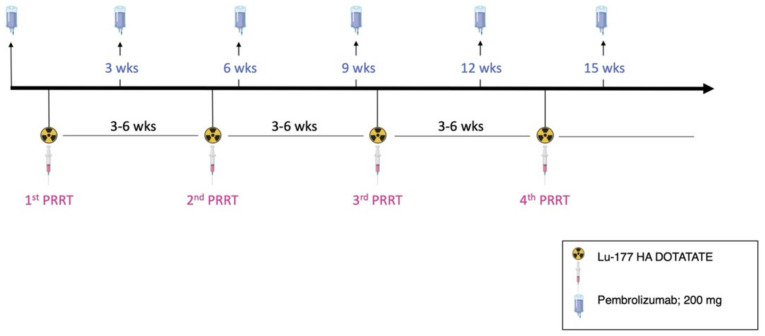
Protocol for the combinational treatment using pembrolizumab and SILD-PRRT.

**Figure 2 pharmaceutics-14-01466-f002:**
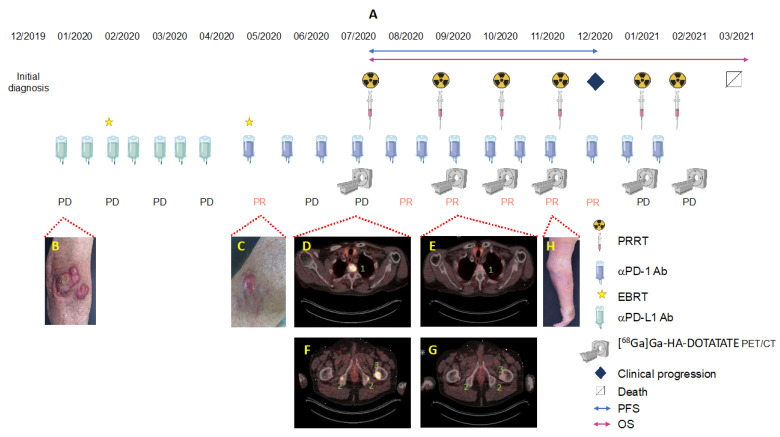
Schematic of the treatment regimen of patient A. (**A**) Clinical course of patient A including photographs of MCC lesions (**B**,**C**,**H**) and fused axial [^68^Ga]Ga-HA-DOTATATE-PET/CT scans (**D**,**F**) before and (**E**,**G**) after 2nd SILD-PRRT with concomitant pembrolizumab. Pathologic foci of enhanced uptake of the radiotracer are indicated by numbers: T2 thoracic vertebral body (1), bone metastases located at the ischial tuberosities (2; both sides), and femur (3; left).

**Figure 3 pharmaceutics-14-01466-f003:**
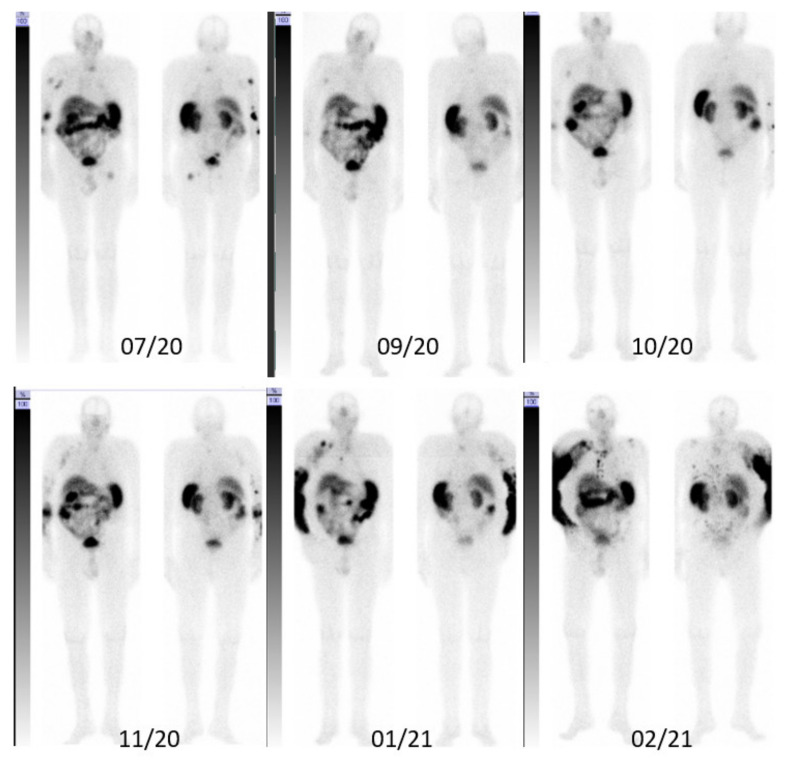
Anterior (**left**) and posterior (**right**) post-therapy scintigraphy images of patient A.

**Figure 4 pharmaceutics-14-01466-f004:**
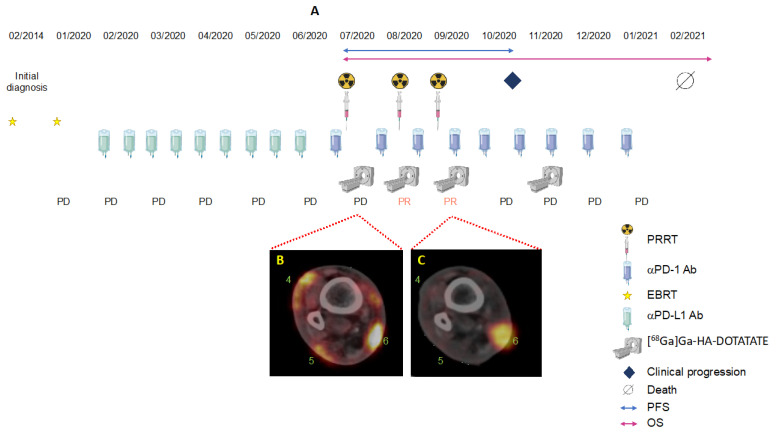
(**A**) Clinical course of patient B with fused axial [^68^Ga]Ga-HA-DOTATATE -PET/CT fused images after the first (**B**) and third SILD-PRRT (**C**). Pathologic foci of enhanced uptake of the radiotracer are indicated by numbers: skin metastases of the lateral (4, 5) and medial lower leg (6).

**Figure 5 pharmaceutics-14-01466-f005:**
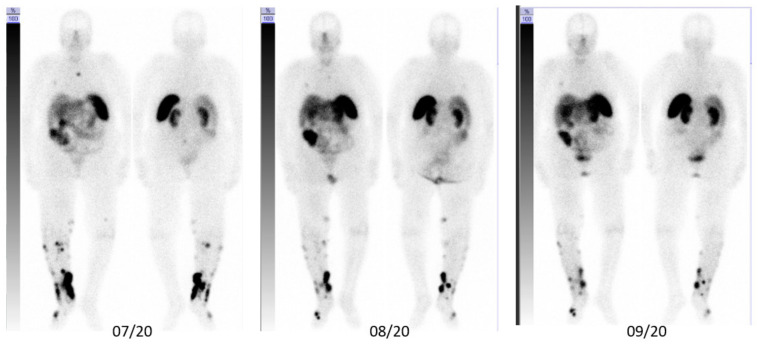
Anterior (**left**) and posterior (**right**) post-therapy scintigraphy images of patient B.

**Figure 6 pharmaceutics-14-01466-f006:**
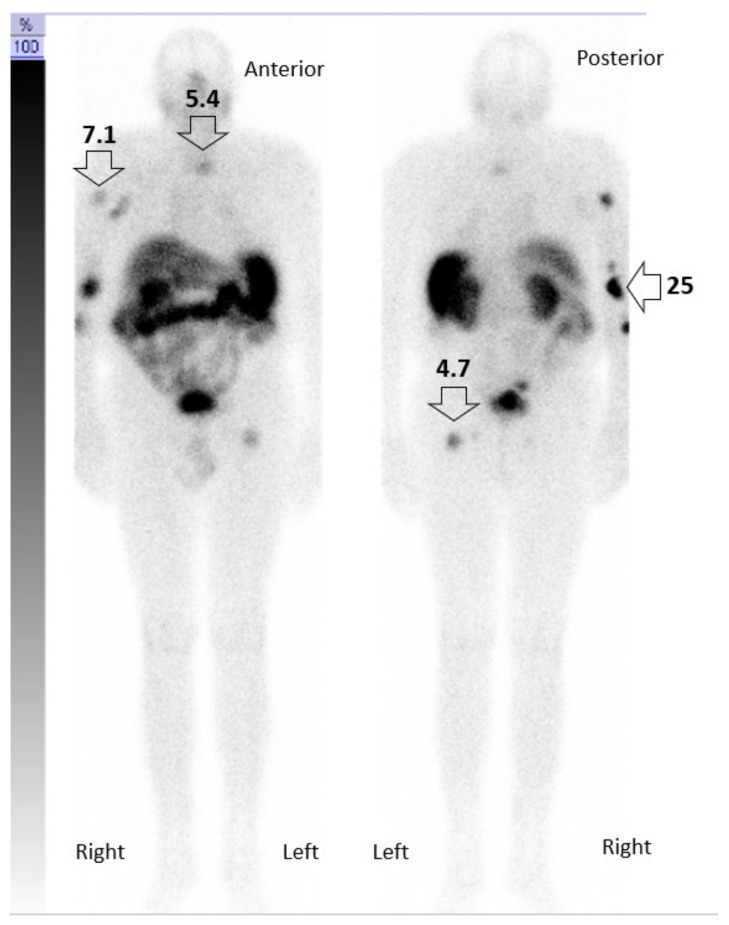
Radiation absorbed dose per PRRT in patient A. Image of the integral activity through time (MBq/min) indicating the local energy deposition (Gy). As an example, the delivered dose in Gy (indicated as numbers above arrows) for one PRRT treatment is indicated.

**Figure 7 pharmaceutics-14-01466-f007:**
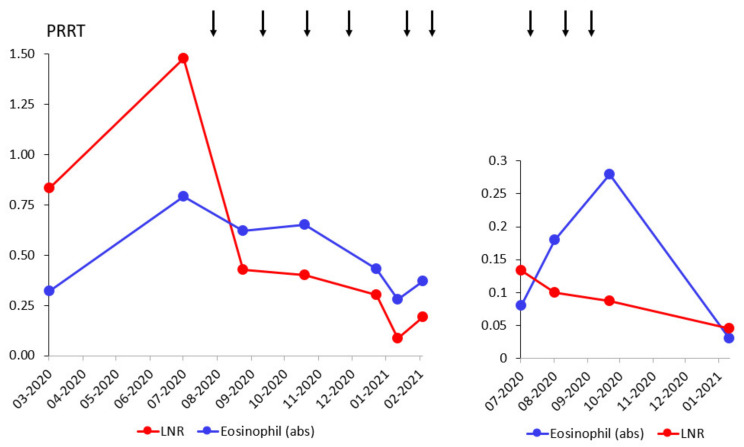
Lymphocyte to neutrophil ratio (LNR), and absolute eosinophil counts (×10^9^/L). Patient A (**left**), patient B (**right**). Arrows indicate the administration of PRRT.

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
