# Peer review of "Short-Interval, Low-Dose Peptide Receptor Radionuclide Therapy in Combination with PD-1 Checkpoint Immunotherapy Induces Remission in Immunocompromised Patients with Metastatic Merkel Cell Carcinoma"

_pharmaceutics, 2022, doi:10.3390/pharmaceutics14071466_

Round 1

Reviewer 1 Report

- In page 2, Figure 2 and in the text above, it calls me the attention that Pembrolizumab (200 mg) treatment was performed regularly (3, 6, 9, 12, 15 wks), while for PPRT therapy it was not the same schedule (time between 1st PRRT and the 2nd one). When PPRT and Pembrolizumab was delivered in the same week, please include the elapsed time between both therapies. This information is missing.

- "During treatment with ICIs, the most common adverse kidney effect is represented by the development of acute kidney injury (AKI) with the acute tubulointerstitial nephritis as recurrent histological feature. The mechanisms involved in ICIs-induced AKI include the re-activation of effector T cells previously stimulated by nephrotoxic drugs (i.e. by antibiotics), the loss of tolerance versus self-renal antigens, the increased PD-L1 expression by tubular cells or the establishment of a pro-inflammatory milieu with the release of self-reactive antibodies" (Front. Immunol., 08 October 2020 | https://doi.org/10.3389/fimmu.2020.574271) https://www.frontiersin.org/articles/10.3389/fimmu.2020.574271/full

Could you please introduce this reference in the discussion? In page 9, it is mentioned: "PRRT combined with ICI was tolerated well with no signs of renal toxicity or generalized hematotoxicity". 

  - J. Nuc. Med. 2022 Jun 9;jnumed.122.263856.doi: 10.2967/jnumed.122.263856. Long-term outcomes of submaximal activities of peptide receptor radionuclide therapy with 177Lu-DOTATATE in neuroendocrine tumour patients. This analysis concludes in a different direction vs your results. However, being your protocol is different, would it be possible to address this issue?5

Reviewer 2 Report

This is a interesting paper about 2 case reports of the association between immunotherapy and PRRT.

- The discussion is well documented and very interesting, and it should be underlined that both patients did not have the same protocol at all.

- The manuscript lacks details about the radiopharmaceuticals infusion : specific activity, volume, aminoacids,... ?

- There is no mention of the regulatory frame allowing these 2 patients to be injected with "homemade" 68Ga-/177Lu-HA-DOTATATE (Committee for person protection ? any regulatory agency ?). If not applicable to your country, please indicate precisely the composition and location of the "multidisciplinary meeting". Was this meeting independant from the team who led the experimentations ?

-P. 3 1st paragraph : please replace 68G-HA-DOTATATE with 68Ga-HA-DOTATATE

- p. 3: ENETS consensus guidelines : please precise which ones and cite as reference

- please write avelumab and not Avelumab, pembro... and not Pembro...

- Fig. 4 : B and C should be resized

Reviewer 3 Report

Although the manuscript was written well with convincing discussion, the proof-of-concept study just enlightened the combination approach efficiency in two patients. As authors mentioned in discussion more patients number and samples along with detailed mechanistic experiments are required to prove this approach.

1.     “By restoring clinical responses in ICI-refractory patients, this combination provides a promising strategy not only for patients with mMCC but many other cancer types currently treated with PD1/PD-L1 inhibitors” . Does the author make this statement based on their research?

2.     Author needs to add in the discussion whether SILD-PRRT can be considered for other ICI based on their experience.

3.     Please include the used Pembrolizumab dose. 
